# Mind the gap: Distributed practice enhances performance in a MOBA game

**Ozan Vardal**[1]*, **Valerio Bonometti**[1], **Anders Drachen**[1,2], **Alex Wade**[3], **Tom Stafford**[4]

**1** Department of Computer Science, University of York, York, United Kingdom, **2** Maersk McKinney-Moeller Institute, University of Southern Denmark, Odense, Denmark, **3** Department of Psychology, University of York, York, United Kingdom, **4** Department of Psychology, University of Sheffield, Sheffield, United Kingdom

* ov525@york.ac.uk

**Data Availability Statement:** The data analysed here are third party data provided by Riot Games, which resemble data available from Riot Games' public API, but also include players' "matchmaking rating" (MMR). The full data set cannot be shared

## Abstract

Understanding how humans master complex skills has the potential for wide-reaching societal benefit. Research has shown that one important aspect of effective skill learning is the temporal distribution of practice episodes (i.e., distributed practice). Using a large observational sample of players ($n$ = 162,417) drawn from a competitive and popular online game (*League of Legends*), we analysed the relationship between practice distribution and performance through time. We compared groups of players who exhibited different play schedules using data slicing and machine learning techniques, to show that players who cluster gameplay into shorter time frames ultimately achieve lower performance levels than those who space their games across longer time windows. Additionally, we found that the timing of intensive play periods does not affect final performance—it is the overall amount of spacing that matters. These results extend some of the key findings in the literature on practice and learning to an ecologically valid environment with huge $n$. We discuss our work in relation to recent studies that have examined practice effects using Big Data and suggest solutions for salient confounds.

## 1 Introduction

Among the many determinants of expertise in skilled human endeavour, the accumulation of experience is one over which the aspiring expert has significant control. Research on skill acquisition and expertise, in particular the framework of "deliberate practice" [1, 2], has demonstrated that the quantity and quality of sustained engagement within a domain of skill is an important driver of ultimate performance. The relationship between practice, performance, and expertise has been subjected to much scientific inquiry [3–7], and despite much debate surrounding its importance in relation to other factors, the effect of practice is widely accepted to be substantial [8, 9]. Researchers seeking to understand and accelerate skill acquisition have adopted a mixture of approaches, including the measurement and comparison of expert and novice performance [10, 11], the tracing of expert thought during practice [12–14], and use of interview methods to elicit expert knowledge [15, 16]. Unfortunately these methods share several difficulties—notably the expenses of recruiting human (expert) samples, the detailed

publicly because of the proprietary nature of Riot Games' matchmaking algorithm and the terms of conditions of data provision. However, the pre-processed data set, which does not include raw MMR values but otherwise permits reproduction of all the present results is available on the OSF website: https://osf.io/hxgtb/.

**Funding:** This work was supported by the EPSRC Centre for Doctoral Training in Intelligent Games & Games Intelligence (IGGI - http://www.iggi.org.uk/) [EP/L015846/1] and the Digital Creativity Labs (digitalcreativity.ac.uk) (JAW, Research Fellow), jointly funded by EPSRC/AHRC/Innovate UK under grant no. EP/M023265/1. The funder had no role in study design, data collection and analysis, decision to publish, or preparation of manuscript.

**Competing interests:** The authors have declared that no competing interests exist.

tracking of cognition and behaviour over periods of training, as well as the use of laboratory tasks that may fail to generalise to the real world.

In recent years, researchers have proposed the use of games as a solution to some aspects of these problems [17–19]. The competitive and immersive nature of many games encourages players to develop profound skill over hours, days and even years of practice. Because most actions taken during a game are recorded on a computer, many competitive online games generate huge reservoirs of ecologically valid performance data that can be requested and interrogated by the curious analyst. Due to their size and richness, naturally occurring data sets from online games afford both statistical power and the ability to extract and examine "participants" that exhibit features of interest to the researcher—features that would usually require experimental manipulation to permit empirical investigation [20]. In the present study we analysed the relationship between skill acquisition and the distribution of practice across time using a data set drawn from *League of Legends*, an immensely popular online game that has previously been estimated to generate over one billion hours of game play per month [21], with a current tournament viewership of over four million spectators [22]. In doing so we generalised a known effect in the psychological literature to a real-world context comprising millions of stakeholders, and extended previous methodological approaches in this space by using clustering techniques to interrogate how learners space their practice sessions across time.

## 1.1 Effects of practice distribution on learning

One aspect of practice that has received considerable attention from researchers is its distribution across time. In the literature on learning and skill acquisition, the effect of distributed practice refers to the tendency of learners to exhibit superior performance following a practice schedule containing rest periods between practice sessions (i.e., distributed practice), compared to a practice schedule containing shorter or no rest periods (i.e., massed practice). The terms distributed and massed practice lack strict definitions—researchers distinguish between the two in terms of the relative amounts of rest time between sessions in different practice schedules [23]. While there is some consensus that distributed practice leads to better learning than massed practice [24–27], it is important to examine what is meant by "learning" in this context, and to consider factors that have been shown to moderate this effect.

The study of distributed practice can be traced back to early studies on the recall of verbal material by Ebbinghaus [28], and so a significant amount of related work has been conducted on the effects of spaced studying on verbal memory, which we will not consider here. However, the effect has also been demonstrated in psychomotor learning [29]. In its simplest form, a study of distributed practice in this context involves participants practicing some motor task (e.g., mirror tracing, rotary pursuit) over a block of practice trials. The amount of rest time between a block of trials (i.e., the "intertrial" or "intersession" interval) in a distributed practice condition is greater than in a massed practice condition, but the spacing between individual trials within each block is kept constant. The researcher then compares performance on a final "test" trial between the two groups. Because learning is said to have occurred when changes in performance are relatively stable [30], more involved designs include a final trial or block of trials separated from the practice block by a non-trivial amount of time ($\geq$ 24 hours). By comparing performance in the "retention" block and the practice block, it can be judged how well acquired performance is retained following a period of no practice. Donovan and Radisevich [25] use the terms acquisition performance (performance in the last trial of the practice block), and retention performance (performance in the first trial of the retention block) to denote this distinction.

Overall, distributed practice appears to have a moderate to large positive effect on motor learning. For example, in a meta-analysis of 47 psychomotor studies, Lee and Genovese [24] reported a large weighted average effect size of.91 for acquisition, and a moderate average effect size of.49 for retention, although the spread on these effect sizes was large. A later meta-analytic review of 61 studies [25] yielded a smaller mean weighted effect size of.46, with a 95% confidence interval ranging from.42 to.50. The authors computed separate averages for effects sizes describing acquisition performance (.45) and retention performance (.51). Noting the importance of the type of task trained in these studies, the authors conducted additional moderator analyses to estimate how task type may influence the magnitude of the distributed practice effect. Ratings of task complexity were collected from 95 graduate and undergraduate students across three dimensions (overall complexity, physical requirements, mental requirements) for all 28 tasks examined in these studies. A cluster analysis resulted in four clusters of task complexity, optimised for maximal within-group homogeneity with meaningful between-group heterogeneity. Correlating between task complexity and effect size suggested that the distributed practice effect is diminished with increasing overall complexity (Pearson's $r = -.25$, $p < 0.05$), while mental and physical requirements were not significantly correlated with the effect. Moreover, bucketing studies into four different levels of intertrial interval, the authors considered the relationship between intertrial interval and task complexity by examining a 4 x 4 matrix of effect sizes. While it was noted that tasks of different complexity may have a different "optimal" intertrial spacing, the observation is caveated by a small number of effect sizes per cell.

## 1.2 Distributed practice in digital games

As mentioned previously, one approach to mitigating difficulties associated with laboratory-based experimentation is through the use of digital games. In cognitive science, a growing body of researchers have advocated for the use of games as an environment for the study of skill learning [17–19] noting several advantages afforded by games that allow researchers to bypass limitations of experimentation. These include large observational data sets (affording statistical power and ecological validity), participants that are intrinsically motivated to engage with the task, and a level of task complexity resembling that of real-world tasks. We review here studies that have used digital games to investigate the spacing effect of practice, in order to provide background on the current work.

Three observational studies of skill acquisition examined the relationship between practice and performance in *Axon*, a casual computer game where players click on periodically generated targets with a mouse to maximise growth of an axon. Performance is measured by a single game score—the final length of the axon. In a first study, Stafford and Dewar [31] analysed digital records of ~850,000 *Axon* players to test the impact of spacing on acquisition. Players were identified heuristically as having distributed (versus massed) their practice if their first 10 plays took place in a >24 hour window (versus <24 hour window). Defined this way, distributed practice had a small but significant effect on subsequent performance (highest score on plays 11 to 15; $d = 0.11$, $p < 0.00001$). Further analysis showed that the association between spacing and acquisition remained after testing separately on subsamples of players with comparable initial ability.

Stafford and Haasnoot [32] extended this work by investigating whether the presence of sleep could explain the effect of distributed practice, and by examining the magnitude of the effect at different levels of spacing. Players in the aforementioned distributed practice category were categorised into a "sleep" or "wake" group based on the timing of their breaks, accounting for geographical location. Comparing average scores between these groups showed no

additional benefit of sleep (in fact, players in the wake group had slightly higher scores than their counterparts). To examine how different rest intervals affected acquisition, the authors plotted average scores of players on plays 11 to 15 against amount of time elapsed between games 1 and 10—an amount ranging from 0 to 60 minutes, discretised into 16 bins. The resulting curve suggested that the relationship between practice distribution and acquisition can be described by a non-monotonic function, where optimal spacing between games lies in the middle of this range.

Agarwal, Burghardt, and Lerman [33] also investigated the relationship between practice and performance by revisiting the *Axon* data set. After segmenting the players' games into sessions (defined as a sequence of games with no longer than 2 hours between consecutive games), they plotted aggregated performance trajectories for sessions of different length (ranging from 4 to 15 games per session), observing that players scored abnormally high on the last game of a session, regardless of session length. Consequently, the authors suggested that the spacing related performance boost observed by Stafford and Dewar [31] could be attributed to this score spike at the last game of a session. The accuracy of this claim is difficult to assess, however, as the two groups of researchers had different quantifications of rest interval, and Agarwal and colleagues did not report any statistics to support this particular observation.

Two studies investigated the effect of distributed practice on acquisition in first-person shooters (FPS), a genre of action video game characterised by fast-paced weapon-based combat in a three-dimensional environment. Importantly, these games are considerably more complex than *Axon* (and many motor tasks employed in the study of distributed practice), seeing as they are played against human or AI opponents, load on bimanual dexterity, and involve communication with other players on the same team. Huang and colleagues [34, 35] reported on the effects of play frequency and breaks between play on performance in *Halo Reach* using a longitudinal data set comprising performance of 3.2 million players over a 7 month period. Players were subsampled by play frequency (operationalised as number of matches played per week), and average performance of each group was plotted first against match, then against time. This produced two perspectives. Players who played a relatively small number of matches per week (4–8) had the fastest acquisition *per match*, while those who played more frequently (>64 matches per week) had the fastest acquisition *over time*. Despite starting lower on initial performance, these players had the highest performance by the end of the 7 month period. These findings show some agreement with the literature on deliberate practice, and illustrate a trade-off inherent to spacing—taking breaks between practice sessions results in greater learning per unit of time invested into practice, but massing of practice can result in the fastest acquisition within a given time period. Additionally, the authors reported a reduction in skill rating following a break from the game longer than a day. However, the magnitude of this reduction grew smaller with an increase in gap size, and in most cases players regained their pre-break skill level after several hours of play. In contrast to Agarwal and colleagues [33], the *Halo Reach* data suggested that players terminate a session of play after a decline in performance rating (associated with a loss) as opposed to after an atypically strong performance.

Stafford and colleagues [36] obtained similar results by observing the performance of players in *Destiny*, another FPS game. Performance was measured by a proprietary "Combat Rating", a Bayesian skill rating system comparable to TrueSkill and Elo [37], systems fundamentally based on a player's win/loss ratio. The authors reported a small but significant positive correlation between performance and distribution of practice (r—0.18, 99% CI [0.14, 0.22]), operationalised as the time range over which players recorded their first 25 days of play. In contrast to results from Huang and colleagues [34, 35], players who spaced their practice started slightly lower on initial ability (Pearson's $r$ = -0.09, 99% CI [-0.14, -0.05]). Additionally, performance over the first 50 matches were plotted for players in the top and bottom quartiles

of spacing, defined as the time gap between the 1st and 25th match. Players who distributed their first 25 matches over a greater time range had higher performance in their subsequent 25 matches. However, this difference was not tested for statistical significance.

Johanson and colleagues [38] are the first group, to our knowledge, to have procured experimental data on distributed practice in digital games. In an online experiment participants played *Super Hexagon*, a minimal action game where players must rotate a triangle inside a hexagon with the aim of avoiding incoming obstacles. Players control the triangle using left and right arrow keys on a keyboard and performance is measured as time until failure. Participants played the game for 5 trials of duration 5 minutes, separated by a rest interval of varying length (5 conditions, ranging from 3 seconds of rest to 1 day). The last trial was a retention test, separated from the preceding trial by one day across all conditions. Analyses revealed a small but significant overall effect of distributing practice on acquisition performance ($\eta^2 = .127$, $p < .001$) and a marginally significant effect on retention performance ($\eta^2 = .108$, $p = .44$). Additional pairwise comparisons showed that practice with no gap resulted in significantly inferior acquisition compared to most conditions. However, the effect on acquisition did not differ significantly between groups with rest intervals, and pairwise differences in retention were not significant at all.

Expanding on this work, Piller and colleagues [39] tested whether the effects of spaced practice are present in a game more complex than *Super Hexagon*, as well as to test differences in acquisition arising from types of break taken. The researchers developed a 2D side-scrolling platformer called *SpeedRunners*, in which players controlled an avatar with the ability to run, jump, and swing with a grappling hook to run laps around a circular obstacle course. Performance was measured as average lap time as well as total distance travelled. Participants played 20 minutes of SpeedRunners split into four 5-minute sessions. Participants in a spaced practice group had breaks of 2 minutes in between sessions, while those in the continuous practice group had 3-second breaks. Participants also returned for a 5-minute test of retention one week after the 20-minute training block. Analyses did not support a positive overall affect of spaced practice on acquisition, but did reveal a small effect of spaced practice on retention performance ($\eta^2 = 0.093$, $p = 0.042$ for average lap time; $\eta^2 = 0.087$, $p = 0.046$ for distance travelled).

## 1.3 Contributions of studies using behavioural telemetry from action games

What do these studies of skill learning in digital games reveal about distributed practice? The reported data are generally in line with previous experiments showing that the cramming of practice into relatively short time frames tends to produce depressed performance following a training period. More specifically, players who distributed their game play sessions over longer time windows exhibited higher performance in subsequent game play sessions, and in some cases during the "training" period itself. In sum, this body of work answers the question as to whether or not practice spacing affects performance, and perhaps learning, in games. Unsurprisingly, it does. Unfortunately, comparing it to previous laboratory experiments of psychomotor tasks is difficult for several reasons.

For one, the majority of these studies were observational in nature, and operationalisations of practice distribution consequently diverged from previous (experimental) approaches. Where in earlier studies practice distribution referred to the amount of time elapsed between individual practice trials or sessions, working definitions in the present studies included the time gap between first and last recorded game instance [31] or game session [36], as well as the number of game instances recorded within a week [34, 35]. Thus, the possible conflation of

practice distribution with practice frequency is a concern. In some cases, data visualisation lacked supporting inferential statistics, making the interpretation of effect significance and size impossible [33–35]. Finally, interpreting players' performance dynamics in commercial games is less straightforward than in laboratory tasks, as performance in the former is typically described by proprietary scoring systems. Taken together, while evincing that the effects of practice distribution persist in complex psychomotor tasks such as action games, the difficulties described above prohibit additional commentary, for instance on the conditions under which the effects might be strongest.

Despite these drawbacks, the studies summarised above highlight several advantages associated with the interrogation of longitudinal, observational data sets. Traditional laboratory experiments of skill acquisition are difficult: Although an observational approach sacrifices experimental control, a sufficiently large data set permits the subsampling of "participants" that meet multiple conditions of interest (e.g., practice at various levels of spacing), and enables the study of skill acquisition over far longer periods than is ordinarily practical (e.g., months). Such data also make it possible to compare the relative impacts of different factors on the dependent variable of interest. For example, Stafford and Haasnoot [32] made an argument for the relevance of distributed practice by demonstrating that the effect of spacing was comparable to tripling the practice amount. In light of these features, the capacity to test theory-led hypotheses using large observational data sets of game performance seems promising.

### 1.4 Aims of the present work

In the current study we extended this line of enquiry to a popular commercial action game, with the aim of generalising work on distributed practice that has been conducted using artificial tasks created by researchers, to a non-artificial, ecologically valid environment with which researchers have not interfered. We analysed a large body of observational performance data to investigate the effects of distributed practice on performance, mirroring operationalisations of practice distribution adopted in recent studies, and extending previous work by using machine learning techniques to investigate how the timing of breaks influences performance gains. In conducting iterative work of this nature, we tested the generalisability of the distributed practice effect in a non-laboratory context comprising millions of stakeholders (e.g., amateur to professional action game players) with a vested interested in fast and efficient acquisition of skill.

## 2 Materials and methods

We used a Python 3.8 [40] environment to preprocess and analyse data, with additional packages for data munging, analysis, and visualisation including Pandas [41, 42], NumPy [43], and SciPy [44]. We used the Pingouin [45] and statsmodels [46] packages for all statistical analyses. All analysis code are publicly available at (https://github.com/ozvar/lol_practice_distribution), together with additional documentation detailing all required software dependencies.

### 2.1 Task environment

Our study focuses on *League of Legends*, a subgenre of action game referred to as Multiplayer Online Battle Arenas (MOBAs). *League of Legends* is one of the most popular competitive online games (esports) in the world, having previously recorded a monthly player base of 67 million players, many of which participate annually in international tournaments [47]. Like other MOBAs, *League of Legends* is a team-based invasion game that involves a high degree of team coordination and fast-paced action as two teams of five seek to destroy the opposing team's headquarters entity, located on the opposite corner of a 2.5D game arena. Each player

uses a keyboard and mouse to control a single game entity (a "champion") selected at the start of each game out of a pool of 150, each with a different set of synergistic combat abilities (e.g., boosting the attributes of friendly champions, immobilising opposing champions). Players must use their abilities to eliminate opponent champions (reanimated after a scaling delay) and computer-controlled entities, as well as to support teammates, in order to reach the win condition of destroying the opposing team's "Nexus". Over the course of the game, each player accumulates "gold" and "experience points" (XP) in proportion to their successful participation in combat with enemies and contest over intermediary map objectives. These resources can be used to strategically modify the abilities and attributes of champions as the game progresses, in order to best adapt to the current game state. The combination of decision making involved in champion selection, modification, and combat, together with the fine motor skills necessary to effectively control champions, makes *League of Legends* a complex game that is hard to master.

Previous studies have used *League of Legends* as an environment to study longitudinal skill acquisition [48], model the relationship between engagement and individual performance in team-based games [49], and investigate teamwork at different temporal resolutions [50–52]. Moreover, as the participation of many players in esports is driven partly by a commitment to skill mastery [53], we anticipate these results to be of interest to relevant stakeholders such as players and professional esports teams, in addition to researchers interested in skill acquisition.

## 2.2 Measures

Whenever players queue online for a match, Riot's servers attempt to balance the teams to ensure a fair game. This balancing is strongly weighted by each player's Match Making Rating (MMR), a relative skill score calculated using a method broadly similar to those used in *Destiny* and *Halo Reach*. That is, a player's rating updates following each match based on the relative skill level of opponents, with wins resulting in an increase and losses a decrease [54]. While MMR is kept hidden from players, it is used to predict a player's ranking in different public tiers and divisions. A player's ranking is visible to other players and determines the skill bracket within which they may play, as well as tournaments that they may qualify for. Thus, while MMR is reflective of skill, individual changes in MMR from match to match may not directly reflect on the performance of any individual player, as MMR is primarily governed by the ratio of wins to losses [54, 55], and the likelihood of a win is dependent on more than the contribution of any single player (e.g., performance of teammates and opponents). For this reason, we concentrated our analyses on post-match statistics that describe the performance of an individual at each match. These included the the amount of gold per minute (GPM) earned in a match, and the ratio of kills and assists scored against opposing champions to the number of deaths experienced by the player's own champion (KDA), calculated using the formula (kills + assists) / max(1, deaths). While metrics like this can be impacted by the role that their chosen champion may fill [56] (e.g., support roles typically earn less gold than the "carry" role), we judged these to be the best available to work with, and had no expectation of systematic bias as players play a variety of roles across their trajectory. As League of Legends developer Riot Games keeps the MMR algorithm confidential, we normalised all values of MMR across the data and analyses reported here.

## 2.3 Data and preprocessing

*League of Legends* developers Riot Games digitally log all match events and summary statistics, and make a subset of all global game logs available to access through a public Application

Programming Interface (API). Presently, we analyse a large data set of game logs describing the longitudinal performance trajectories of *League of Legends* players across matches. Our data closely resemble that which is available through the API, but were provided to us by Riot Games and therefore differ in that they additionally contain a record of player MMR at each match, which is ordinarily not publicly available. The data comprise all ranked matches played by a random sample of 482,415 new *League of Legends* accounts over the course of a competitive season, dating from 21 January 2016 to November 2016. All analyses were in compliance with the terms and conditions for data usage made clear to us by Riot Games. All matches correspond to the default "Solo/Duo Queue" ranked mode of play, with five players on each team. Each row in the data lists a single match for a single given player, containing a unique player identification number, unix timestamp, and various performance and outcome variables (see Table 1 for an overview of the raw data). Importantly, these were newly created accounts that had not previously been registered with any competitive *League of Legends* play prior to the start of this season. New player accounts are initialised at the same MMR value when they first enter ranked play, and therefore nominally appear to be of equal skill at the start of their trajectories. However, as the data set lacks records of unranked matches that may have been played in order to unlock the ranked game mode, we are limited in our knowledge of differences arising from prior experience. Additionally, as all account IDs are anonymised, we cannot associate each ID with a single unique player, and acknowledge hereby another source of potential bias, although we do not expect it to be systematic.

We took several steps to ensure the quality of the data prior to analysis. These preprocessing steps were focused on ensuring data quality for an initial window of 100 matches, as visualisation indicated that this was the period in which most players appeared to reach asymptotic performance. We first dropped all players who had not played a minimum of 100 games over the course of the season, and any players with missing values in any of their first 100 match records. We dropped any players who had a non-default initial MMR value, as well as players with records in multiple servers, as these observations violate our assumption of equal starting

**Table 1. Raw data columns available in a single row of the data set analysed in this study.**

| Column | Description |
|---|---|
| Account ID | Unique anonymised numeric identifier of player account |
| Platform ID | Identifier of server the match was played on |
| Game ID | Unique numeric identifier of match |
| Neutral Creep | Number of neutral AI entities killed |
| Enemy Creep | Number of opponent AI entities killed |
| Win | Boolean indicator of match result |
| Timestamp | Unix timestamp indicating when the match was logged |
| Date | Date on which the match was played |
| Hour | Hour at which the match was played |
| Gold Earned | Total amount of Gold earned by the player |
| Damage Dealt | Total damage dealt by the player to opponents |
| Time Dead | Total time in seconds the player champion spent dead |
| Time Played | Total time in seconds played in the match |
| Kills | Total kills scored on opponent champions |
| Deaths | Total number of times the player champion was killed |
| Assists | Total number of times the player assisted in scoring a kill |
| Rating | Normalised MMR of the player before the match |
| Position | Role of the player champion |

experience. These inconsistencies can occur when a player migrates from one server to another, and would have confounded our assumption that all accounts in the sample started with similar experience. We also dropped any players with matches that lasted less than 900 seconds within the first 100 matches we sampled, as this is indicative of a match which has been abandoned by one or more players, and thus does not reflect a match experience that is on equal terms with all others in the sample. Finally, we removed any players with games in which they were likely completely inactive (i.e., matches in which they scored 0 Kills, Assists, Deaths, and Creep Kills). In addition to dropping players that did not meet analysis requirements, we performed several linear combinations of columns from the raw data to generate additional variables of interest: KDA, GPM, and the time gap between the end of one match and the start of the next. We retained a total of 162,417 players following preprocessing and a corresponding 16,241,700 rows worth of data (at 100 matches per player).

## 3 Results

To assess general changes in performance as a function of experience, we first plotted the trajectory of GPM and KDA against matches played for all players in the sample (Fig 1). The trajectories of average GPM and KDA per match displayed a sharp initial climb with decelerating gains. This is in line with previous studies that have found good fit between the power or exponential function and averaged performance, demonstrating the diminishing returns of sustained experience on performance across a range of domains [57–59]. We also plotted the averaged MMR trajectory of all players in the sample which, in contrast, sharply decreased before showing a gradual rise towards later matches (S1 Fig). We attributed this initial rating drop to our sample being composed exclusively of new accounts. Specifically, we expected new players to suffer more losses against the relatively more experienced majority (unobserved in the sample) towards the start of the season, where the matchmaking algorithm has begun to calibrate for fair matches. This intuition is supported by the trajectory of loss percentage, which descends to 50% as the average rating of the sample stabilises (plotted together with MMR).

We assessed the effects of spacing on acquisition performance first by subsampling and comparing groups of players with different patterns of spacing. We concentrated these

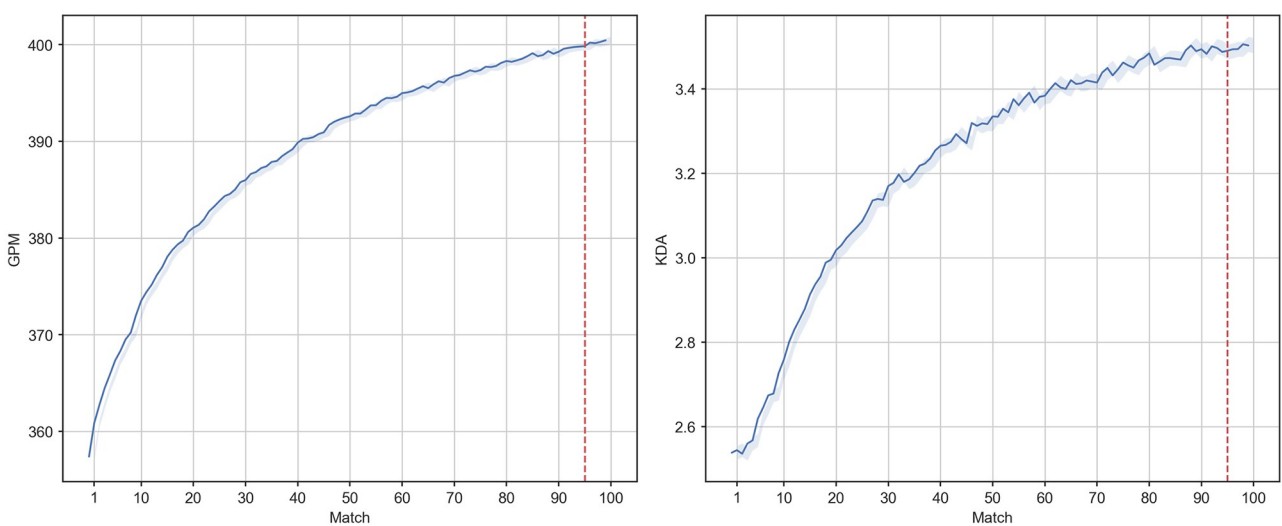

**Fig 1. Trajectories of mean GPM (left panel), and KDA (right panel) of all players against match.** Shaded regions indicate 95% confidence intervals.

analyses on the first 100 matches, as player performance appeared to asymptote towards the end of this window, and we were predominantly interested in acquisition effects. Similar to Stafford and colleagues [36] spacing was operationalised as the gap in days between the 1st and 95th game. After visualising the frequency distribution of time in days elapsed between the first and 95 match for each player (S2 Fig), we subsampled three groups of players that were sufficiently discrete in terms of their break schedules, and that were adequately sized for statistical analysis: players that took between 136–150 days, 76–90 days, or up to 15 days to play their first 95 matches. Visualising the impact of gap size on mean performance over the final five (96th to 100th) matches, we initially observed that while players who spaced their first 95 matches over a greater range had higher acquisition, players who massed their matches in a shorter range initialised at much higher initial performance (close to the maximum observed performance). Due to the negative correlation between this time range and initial GPM (Pearson's $r$ = -0.295, 95% CI [-0.30, -0.29]), we suspected our spacing measure to be confounded by initial performance, potentially explained by a combination of play intensity and other factors related to ability.

In order to control for initial levels of absolute performance, we subsampled players who scored a mean GPM of between 315 and 385 (an interval centered on the median of mean initial GPM; 350 ± 25) over their first five matches, resulting in a subsample of size $n$ = 52,440. Analogously, we replotted KDA trajectories after subsampling players with a mean KDA of between 1.64 and 2.24 (median initial KDA 1.94 ± 0.19), resulting in a subsample of size $n$ = 17125. Fig 2 shows the mean GPM and KDA trajectories of players who took between 136–150 days, 76–90 days, or up to 15 days respectively to play their first 95 rated matches. Players who clustered their matches the most exhibited a faster initial climb in initial, but lower performance overall by the end of their trajectory. Although we produced an analogous plot for mean trajecotires of MMR (S3 Fig), we neglected to conduct further (statistical) analyses of this metric due to the aforementioned opaqueness of the MMR algorithm and the ubiquitous downward trend in MMR across our entire sample, which we believe lent itself poorly to a study of learning.

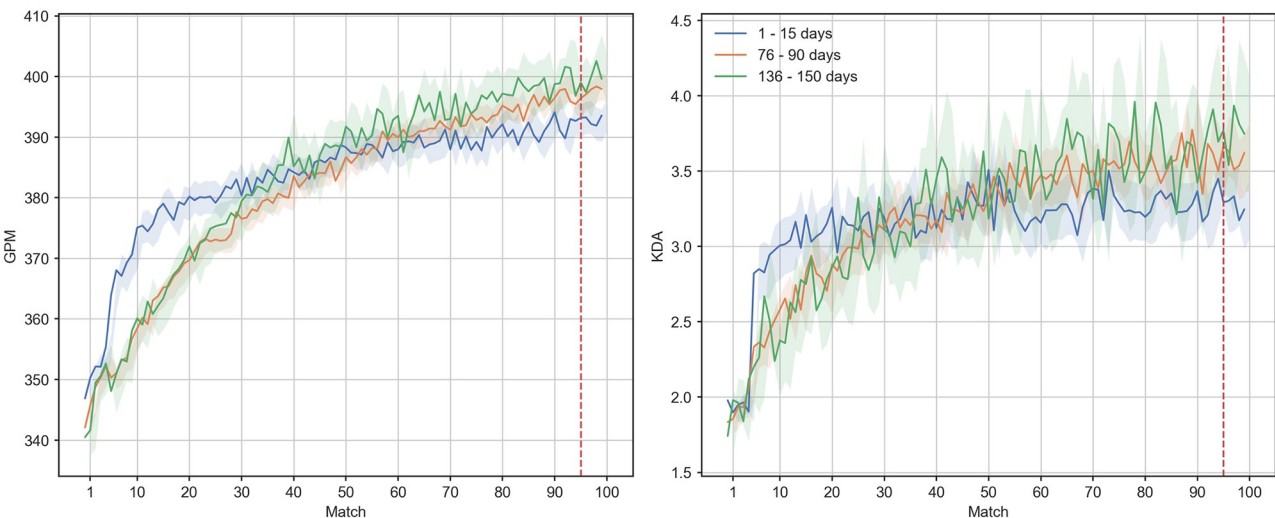

**Fig 2. Trajectories of mean GPM (left panel) and KDA (right panel) against match for players with different patterns of match spacing.** Data in the figure are a subsample of players who initiate at a similar range of GPM and KDA (approximately surrounding the original sample median). Shaded regions indicate 95% confidence intervals.

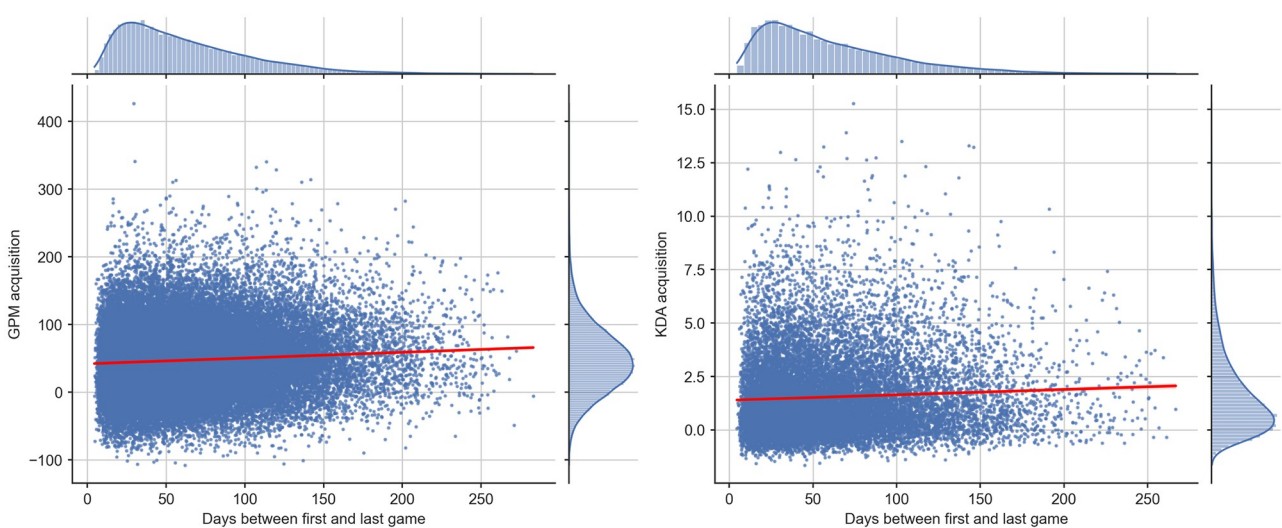

**Fig 3. Scatter plots of GPM and KDA against time range in days between first and 95th game respectively, with line of best fit.** Axis plots show distributions of respective axis variables.

Players with the largest time range between their 1st and 95th match achieved an average GPM in their final five matches that was 6.91 points higher (95% CI [3.74, 10.07], $n = 1236$, $M = 399.71$, $SD = 49.65$) compared to those with the smallest time range (i.e., 1–15 days; $n = 2790$, $M = 392.81$, $SD = 46.18$). This was statistically significant following a t-test at $t(4024) = 4.28$, $p < 0.001$, albeit for a small effect size (Cohen's $d = 0.146$). For the subsample matched on initial-KDA, players in the former ($n = 373$, $M = 3.76$, $SD = 2.18$) achieved a KDA 0.49 points higher (95% CI [0.28, 0.71]) points higher that those in the latter spacing group ($n = 1159$, $M = 3.27$, $SD = 1.74$) This difference was also statistically significant [$t(1530) = 4.45$, $p < 0.001$, $d = 0.265$].

By binning players using our spacing measure, we produced a snapshot of the effects of practice distribution on performance. To produce a fuller account of this relationship using the entire range of our practice distribution variable, we linearly regressed spacing on both GPM and KDA (Fig 3). We report regression slopes and supporting statistics for both variables in Table 2. We report White's heteroscedasticity-consistent standard errors [60] due to non-constant variance in our residuals.

## 3.1 Time gap clustering

One issue with operationalising practice distribution as the time range between two matches, is that different schedules of practice may coexist within identical time ranges. For instance, a player with a consistent schedule of 1–2 matches per day could be grouped with a player who played 10 matches per day followed by a handful of matches after a 10 week break. To explore whether our spacing groups reflected the differences in practice distribution that we were interested in, as opposed to some other systematic and unanticipated differences in play

**Table 2. Linear regressions of time delta in days between 1st and 95th match on average GPM and KDA between the 96th and 100th match.**

|  | B | Std. Err. | β | T | p | R² | 95% CI |
|---|---|---|---|---|---|---|---|
| GPM | 0.0849 | 0.005 | 0.0727 | 16.045 | <0.001 | 0.005 | [0.075, 0.095] |
| KDA | 0.0025 | <0.001 | 0.0568 | 7.396 | <0.001 | 0.003 | [0.002, 0.003] |

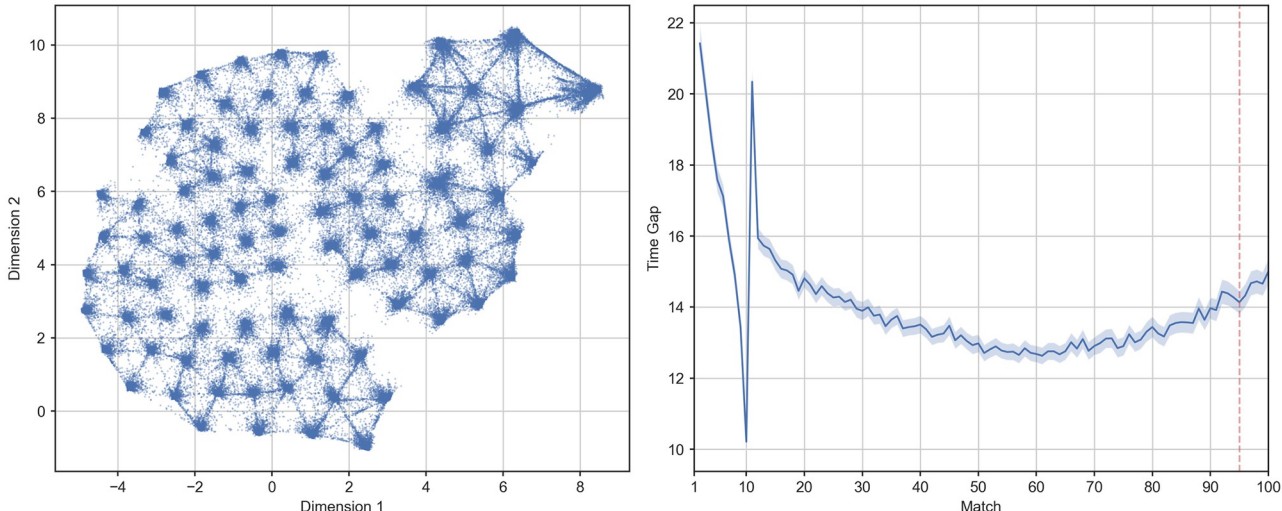

**Fig 4. The left panel shows the two-dimensional projection of the observed 95 inter-match gaps in hours as generated by UMAP for the entire sample**. The y and x axes represent the two dimensions individuated by UMAP. As opposed to Principal Component Analysis their associated values should be interpreted as coordinates on a plane rather than indicators of the magnitude of the two components. Each dot represents the history of inter-matches gaps for a single player while distance between dots indicates the degree of similarity between different patterns of spacing. **The right panel shows the average evolution of inter-match gap in hours for the entire sample**. The y axis indicates the time in hours elapsed since the previous match while the x axis indicates the order of the match. The solid line indicates the mean value while the shaded region shows the 95% confidence interval. The dotted red line separates the observation period (i.e., the first 95 matches) from the evaluation period (i.e., the last 5 matches).

schedules, we conducted an alternative analysis to the rule-based slicing performed above by clustering our original sample of 162,417 players by their time series of time gaps between matches. First, we leveraged the Uniform Manifold Approximation and Projection (UMAP) algorithm [61] to perform a visual inspection of how different players distributed their matches over time. The UMAP algorithm is a non-linear dimensionality reduction technique based on manifold learning. Given a high dimensional data-set, UMAP first infers its topological structure and then using stochastic gradient descent attempts to structurally reproduce it in a lower dimensional space (two or three for visualization purposes). In our case, the original data-set was represented by an $N \times T$ matrix of between matches time gaps, with $N = 162,417$ being the number of considered players and $T = 95$ the number of matches in the observation period. We chose this range to align with the previous step of our analysis, allowing a window of five final matches with which to analyse the effects of different spacing patterns on final performance. The transformation performed by UMAP generated an $N \times D$ matrix with $D = 2$ being the number of target dimensions. In this 2D representation, players with a similar pattern of inter-matches temporal gaps were represented closer in space while players with a dissimilar spacing profile were represented as far apart. The topological structure of the original data-set was inferred by computing the euclidean distance in a local neighborhood of 1000 points, while the dimensionality reduction was achieved by running the optimization part of the algorithm for 1000 iterations. The remaining parameters were left at their default value as provided by the python library used for our analysis (i.e., UMAP-learn [61]). The generated 2D representation can be observed in Fig 4.

As we can observe form Fig 4, a number of naturally occurring groups appear to emerge (i.e., the areas where the density of dots increases), suggesting the existence of different profiles of play distribution. In order to formally evaluate whether differences in naturally occurring spacing patterns truly exist, we decided to run a clustering analysis, adopting three different approaches. This was done to test the consistency of the individuated profiles arising from

clustering. Due to space constraints we will only describe and report the results derived from a combination of recurrent autoencoder and mini-batch K-means. Details and results for the remaining two approaches can be found in S1 Appendix, together with S5–S9 Figs.

**Recurrent autoencoder and K-means.** Autoencoders are a specific type of artificial neural network (ANN), which given an input $x$ attempt to produce a copy of the same [62]. This is done by simultaneously learning the parameters of a function $h = f(x)$ (called encoder), mapping the original input to a latent representation $h$, and of a second function $\hat{x} = g(h)$ (called decoder), generating a copy $\hat{x}$ from the same latent representation [62]. Learning occurs through stochastic gradient descent, minimizing a reconstruction loss that measures the mismatch between $x$ and $\hat{x}$. Once the training process is terminated the latent representation $h$ can be extracted, and should carry meaningful properties of the original input. In this sense, the operations performed by the encoder function can be seen as a form of automatic feature extraction.

In order to force the autoencoder to produce an $h$ with interesting characteristics, a series of constraints are usually applied during the learning process. In our case we adopted a combination of denoising and undercompleteness strategies. The first corrupts the input (usually through random gaussian noise) forcing the autoencoder to learn a representation capable of undoing the noise, while the second requires the dimensionality of $h$ to be much smaller than that of the original input [62]. Since we were dealing with time-series data, we parameterized the encoder and decoder functions using two recurrent neural networks (RNN), a specific type of ANN able to capture temporal dynamics [62]. The first RNN tasked to generated $h$, was composed of two Long Short-Term Memory (LSTM) [63] layers respectively with 60 and 30 hidden units. The second RNN, used to reconstruct the corrupted input was a single LSTM layer with 60 hidden units. The autoencoder minimized the Mean Absolute Error (MAE) between the reconstructed and original inputs and used the Adaptive Moment Estimation (Adam) optimizer [64] for gradient descent. Training was carried out by passing random batches of 512 inputs and monitoring the reduction in MAE on a 20% held-out subset of the original data. Training was terminated once the reconstruction loss stopped decreasing in the held-out subset by a minimum of $\delta = 0.0001$ for more than 15 consecutive epochs. At this point, we proceeded to generate features from the original input passing a $N \times T \times 1$ tensor of between matches gaps through all the operations carried out by the encoder function. This generated an $N \times h$ matrix, with $h = 30$ being the dimensionality of the last layer of the encoder, which other than offering a more compact representation of the original input (making it easier to perform a cluster analysis) should have also distilled its most salient characteristics.

Finally, in order to obtain different spacing profiles we applied Mini Batch K-Means (a more scalable version of K-Means) [65] to the representation generated by the encoder. We selected the number of centroids $k$ by generating an elbow plot after running the algorithm for a range of 2 to 10 $k$, with 2000 random initializations for a maximum of 3000 iterations each, passing the inputs in random batches of 512 elements. Following the methodology proposed by Satopa et al. [66], the optimal $k = 4$ was found by individuating the point of maximum curvature in the aforementioned elbow plot (S4 Fig). In order to derive interpretable profiles from the individuated cluster, we averaged the time series of between-match time gaps (along with GPM and KDA) over the labels provided by the Mini Batch K-Means. The autoencoder was realized using tensorflow's high level API keras [67, 68], while the Mini Batch K-Means implementation we employed was the one provided by the library scikit-learn [69]. Results of this clustering analysis can be seen in Figs 5 and 6.

Looking at Fig 5 we can see how the location and extension of the clusters on the 2D reduction provided by UMAP tells us when, for how long and how intensely the players in those

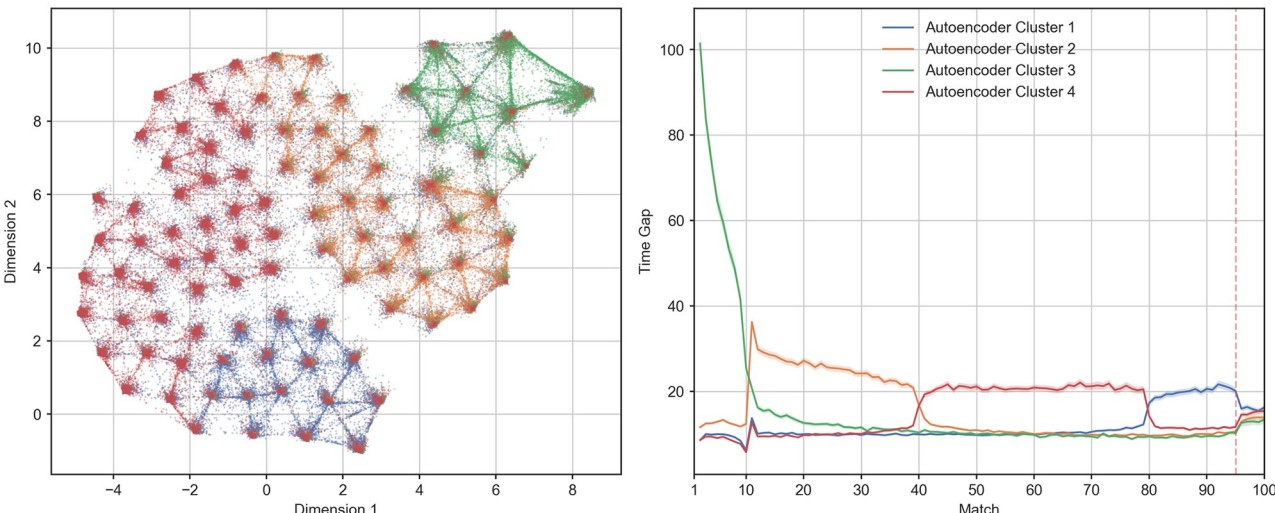

**Fig 5. The left panel shows the two-dimensional projection of the observed 95 inter-match gaps in hours as generated by UMAP for each spacing cluster across the entire sample**. The y and x axes represent the two dimensions individuated by UMAP. As opposed to Principal Component Analysis their associated values should be interpreted as coordinates on a plane rather than indicators of the magnitude of the two components. Each dot represents the history of inter-match gaps in hours for a single player while distance between dots indicates the degree of similarity between different patterns of spacing. **The right panel shows the average evolution of inter-match gap in hours for players in each spacing cluster**. The y axis indicates the time in hours elapsed since the previous match while the x axis indicates the order of the match. The solid line indicates the mean value while the shaded region shows the 95% confidence interval. The dotted red line separates the observation period (i.e., the first 95 matches) from the evaluation period (i.e., the last 5 matches).

clusters spaced their matches on average. Interestingly, the areas of high density in this representation seem to identify groups of players taking a single long break at specific points during our observation period. With the exception of a single period characterised by longer breaks (more hours) between matches, players appear to maintain a consistent play schedule. Following the representation in the right panel of Fig 5 we can see that clusters 1 and 3 represent the

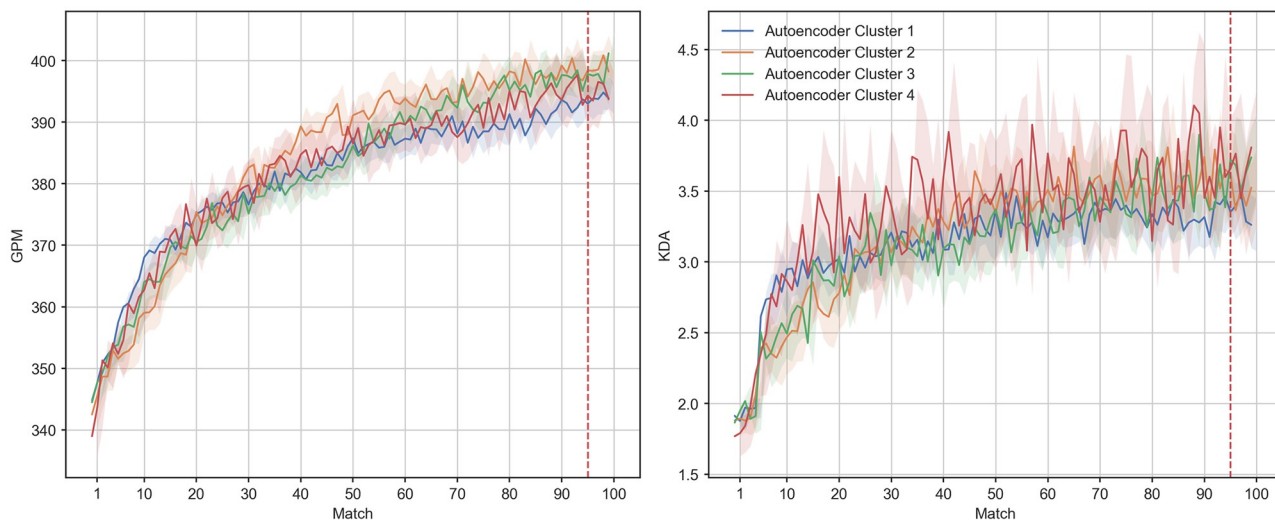

**Fig 6. Trajectories of mean GPM (left panel) and KDA (right panel) against match for players in our 4 autoencoder clusters.** Data in the figure are a subsample of players who intiate at a similar range of GPM and KDA (approximately surrounding the original sample median). Shaded regions indicate 95% confidence intervals.

**Table 3. Joint frequencies of players in spacing group as defined by k-means cluster (rows) versus time in days delta between 1st and 95th match (columns).**

|  | 1–15 Days | 76–90 Days | 136–150 Days |
|---|---|---|---|
| K-means cluster 1 | 7701 | 3477 | 914 |
| K-means cluster 2 | 1644 | 4120 | 1102 |
| K-means cluster 3 | 583 | 2093 | 523 |
| K-means cluster 4 | 2014 | 3430 | 1006 |

extremes of a continuum going from a relatively early versus late rest period. Clusters also differed on the intensity of this rest period, with spacing cluster 3 exhibiting the longest breaks during the shortest rest period, followed by the most consistent streak of play. These results, albeit with some variation, appear to be consistent across all the clustering methods (see S5–S8 Figs).

Tabulating the joint frequencies of players across each of the clusters and original categories (Table 3) showed that players in a given spacing category do not display uniform membership to a single spacing cluster, supporting our intuition that operationalising practice distribution as a time range may mask differences in underlying play schedules.

Fig 6 displays the typical averaged trajectories of GPM and KDA for players in each cluster. Compared to the analysis of groups sliced by time range, there were no large differences between spacing clusters in final GPM or KDA. We conducted one-way ANOVAs to test these differences in mean final performance (average GPM and KDA over the last 5 matches). This was significant for both GPM [$F(3, 162413) = 517.93$, $p < 0.001$] and KDA, [$F(3, 162413) = 439.87$, $p < 0.001$] but for negligible effect sizes ($\eta^2 < 0.01$). Additionally, we conducted pairwise comparisons (Holm-Bonferroni corrected t-tests) in GPM and KDA between each pair of clusters. We identified significant differences in GPM between clusters 1 and 2 [$t(31839) = 5.45$, $p < 0.001$, $d = 0.061$], clusters 1 and 3 [$t(24758) = 3.65$, $p < 0.001$, $d = 0.051$], as well as clusters 3 and 4 [$t(30911) = 4.23$, $p < 0.001$, $d = 0.049$], but only negligible effect sizes. Clusters 1 and 3 were also significantly difference in mean final KDA [$t(8195) = 3.05$, $p < 0.001$, $d = 0.075$], but again with a negligible effect size.

## 4 Discussion

Analysing a large data set drawn from *League of Legends*—one of the world's most popular competitive online games—we extended recent work on the distributed practice effect [31, 32, 34–36] in an ecologically valid and complex perceptual-motor skill environment. Players in our data set showed monotonic gains in measures of absolute performance (GPM, KDA), which tapered off after approximately 100 matches. After matching players on initial ability and subsampling groups defined by the amount of time elapsed between their 1st and 95th game, we found that players who spaced practice the most showed initially depressed gains but superior final performance, albeit for a small effect size, and only for a large time range of spacing. These effect sizes were in line with those previously reported in action video games [31, 36, 38]. In a second analysis, we applied clustering techniques to identify and analyse differences in the *timing* of practice spacing in our data set, and tested whether the "when" of practice distribution has an effect on performance. Our analyses indicated that, for this task environment, only the total amount of rest is what matters, and not the timing of these rest periods. Practically speaking, our results suggest that by their 100th match, a player who maximised spacing would be earning on average 228 gold more and scoring a KDA of 0.49 higher per match than a player who crammed their matches, given the typical match lasted roughly

33 minutes in our sample. Although highly significant, our effects are limited by large spread around our group means. This observation echoes concerns raised in recent research, namely, that analyses of aggregated data sacrifice the ability to accurately describe dynamics of the individual [18].

For the sake of completeness, we also reported players' trajectories of MMR, a relative measure of performance calculated by a proprietary algorithm that is heavily weighted by match outcome (i.e., win versus loss; [37, 54, 55]). A full description of the MMR algorithm is kept hidden from the public, making MMR significantly more opaque than GPM or KDA as a measure of performance. Moreover, although MMR is partly dependent on match outcome, the probability of winning a match is dependent on many factors (including the behaviour of teammates and opponents), and is itself the subject of many efforts in prediction. For these reasons we neglected to conduct further statistical analyses of MMR, and instead concentrated our efforts on GPM and KDA, which we believe to provide a clearer perspective on individual performance from match to match.

The size of our effects (Cohen's $d$ = 0.146 for GPM; 0.265 for KDA) are in keeping with other studies of digital games that reported on the distributed practice effect. For example, Stafford and colleagues reported a small effect size of distributed practice on subsequent performance in *Axon* (Cohen's $d$ = 0.11; [31]), a small correlation of distributed practice on the slope of performance in *Destiny* (Pearson's $r$ = 0.18; [36]), while Johanson and colleagues [38] reported a small effect of distributed practice on acquisition ($\eta^2$ = .127, $p$ <.001) as well as a marginally significant effect on retention ($\eta^2$ = .108, $p$ = 0.44). Importantly, it is also consistent with early meta-analytic work that observed smaller effect sizes in studies involving motor tasks of lower overall complexity [25]. Despite efforts to mimic related work, we are cautious to make direct comparisons between the effects reported here and similar studies due to differences in elements of study design, such as the length of our training window and our operationalisation of practice distribution, as well as the exploratory nature of our design. An explanation as to why practice distribution is less beneficial for more complex tasks presumably depends on a fuller understanding of the mechanisms underlying memory consolidation and the effects of extended inactivity on subsequent recall. Ultimately this is a question for future experimental work that investigates the effects of distributed practice while directly manipulating levels of task complexity.

Our initial results appeared to be confounded by pre-existing differences in gameplay habits. Similar to Stafford and colleagues [36], the distribution of practice was significantly related to the intercept of performance in our sample, but to a more extreme degree. Specifically, players who clustered their matches in relatively shorter time windows initiated at much higher levels of absolute performance. Plausibly, we were observing in our "groupers" a category of player characterised by intense, frequent play. Such players may be more motivated to engage with the game, and would potentially have accrued a commensurately higher amount of experience during the early initiation period of the game where only unranked matches can be played. We attempted to control for this confound by running our analysis on a subsample of players matched on initial performance, but acknowledge that lingering effects of this confound may nonetheless impact our reported statistics.

Similarly, as our sample consisted only of ranked matches, we were agnostic to any experience that players may have acquired in unranked matches that were played between the ranked matches recorded in our data set. A related concern is that players we found to have spaced their matches the most may have played more matches generally than players in our massed practice group, having had more opportunities to play unranked matches during breaks from the ranked game mode. However, we contend that our observations are inconsistent with this hypothesis, as we would then have expected the players that spaced their matches the most to have a more

accelerated learning trajectory than what is observed in Fig 2, reflecting the additional practice hours that they accumulated. Nevertheless, we suggest it is important for related future work to eliminate any such ambiguity by ensuring that the entire history of player experience is visible when curating the data. In this regard, it may be also be worthwhile to record players' past experience with other digital games. In their analysis of gameplay patterns in *Halo Reach*, Huang and colleagues [34] reported separate rating trajectories for players that had previous experience in various related games, such as previous iterations of the Halo series, or other FPS games. This showed that differences in prior experience resulted in differences in current rating. Thus, we suggest that future work could deliver more precise results by capturing pre-existing differences in game experience, for instance through an additional survey component.

Previous work that has leveraged game telemetry data to study distributed practice in games has made use of data slicing techniques to isolate play schedules of interest [31–33, 36]. As an extension to this approach, we used machine learning to cluster players by their time series of gap between matches. In doing so we aimed to reveal naturally occurring play schedules in our data set and investigate whether these underlying patterns have any bearing on effects arising from our data slicing procedures. Our results showed that players in the same spacing group, defined by the time delta between two matches, may diverge considerably in their underlying play schedules, as identified by our time series clustering. Specifically, players across different spacing clusters differed in the timing of an extended "rest" period, characterised by less frequent gameplay. This suggests that operationalising practice distribution as a time delta between two matches may not be as straightforward an analogue to classical operationalisations as one would have hoped. Nonetheless, players across these spacing clusters did not differ significantly in their final performance, suggesting that it is indeed the amount of time spent on breaks that impacts acquisition, but not necessarily the timing of these breaks.

By identifying and attempting to control for confounds in our data, we highlight both a weakness and a corresponding strength of telemetry-based big data analysis. The use of observational data in behavioural science sacrifices total control of participant behaviour. In our case, the absence of experimental control restricted our ability to compare groups of players with homogenous time gaps between each of their play sessions, as has been done in laboratory studies of distributed practice [23]. Our solution, similar to other studies that have used game telemetry [31, 32, 36] was to use a proxy for intersession time interval, namely the time gap between the first and last match. Although time between first and last match is likely related to time between individual trials, we acknowledge that use of this alternative operationalisation limits our ability to generalise from laboratory work to a non-artificial environment.

An additional consequence of using observational data is susceptibility to the effects of both known and unknown nuisance variables that may systematically skew results in unpredictable ways. Presently we attempted to filter out potential confounds, such as players that migrated server (accumulating additional "hidden" experience), or players whose records contained matches with abnormal participation (i.e., complete inactivity). In doing so we dropped approximately two thirds of our data, but were nonetheless left with a sample size that offered ample statistical power. However, despite our attempts to isolate our variables of interest, we remain cognizant of the potential for additional confounding variables. These may include the presence of multiple players using the same *League of Legends* account, or the existence of highly experienced players who create new accounts to enjoy lower levels of ranked play ("smurfs").

## 4.1 Conclusion

Research on motor learning has demonstrated that taking breaks between practice sessions, as opposed to massing them in relatively short time windows, benefits ultimate performance [24,

25]. By analysing an observational, longitudinal data set describing player performance in a massive, commercially successful video game, we showed that the distributed practice effect is relevant in an ecologically valid context comprising stakeholders with a vested interested in improving their skills. Although data sets such as ours afford strong statistical power and the ability to filter through observations that meet desired experimental conditions, they are also complicated by noise and potential confounds. As a solution, we propose that researchers seeking to use telemetry data adopt a hybrid approach, collecting demographic information on players before tracking their play records through game APIs. In doing so, interested researchers may control for variables related to initial performance, such as age or cognitive ability [70], and account for sources of data pollution such as players generating data on multiple accounts.

## Supporting information

**S1 Appendix.**
(DOCX)

**S1 Fig. Trajectories of mean GPM, normalised MMR, and loss percentage.**
(PNG)

**S2 Fig. Histogram of hours elapsed between first and last match played for each player.**
(PNG)

**S3 Fig. Trajectories of mean normalised MMR (left panel) and GPM (right panel) for three groups of players with different patterns of match spacing.** Players in this figure are a subsample who initiate at a similar range of GPM and KDA respectively (approximately surrounding the original sample median). Shaded regions indicate 95% confidence intervals.
(PNG)

**S4 Fig. Elbow plot of inertia against number *k* of centroids while running the Mini Batch K-means clustering algorithm.** The dotted vertical red line indicates the point of maximum curvature and thus the selected number of optimal *k* clusters for our K-means clustering of gameplay schedules.
(PNG)

**S5 Fig. The left panel shows the two-dimensional projection of the observed 95 inter-match gaps in hours as generated by UMAP for each K-means cluster of play scheduling.** The y and x axes represent the two dimensions individuated by UMAP. The associated values should be interpreted as coordinates on a plane rather than indicators of the magnitude of the two components. Each dot represents the history of inter-match gaps in hours for a single player while distance between dots indicate the degree of similarity between different patterns of spacing. The right panel shows the average evolution of inter-match gap in hours for each K-means Cluster. The y axis indicates the time in hours elapsed since the previous match while the x axis indicates the order of the match. The solid line indicates the mean value while shaded regions show 95% confidence intervals of the mean. The dotted red line separates the observation period (i.e., the first 95 matches) from the evaluation period (i.e., the last 5 matches).
(PNG)

**S6 Fig. The left panel shows the two-dimensional projection of the observed 95 inter-match gaps in hours as generated by UMAP for each cluster of play scheduling identified via HDBSCAN.** The y and x axes represent the two dimensions individuated by UMAP. The

associated values should be interpreted as coordinates on a plane rather than indicators of the magnitude of the two components. Each dot represents the history of inter-match gaps in hours for a single player while distance between dots indicate the degree of similarity between different patterns of spacing. The right panel shows the average evolution of inter-match gap in hours for each density-based cluster. The y axis indicates the time in hours elapsed since the previous match while the x axis indicates the order of the match. The solid line indicates the mean value while shaded regions show 95% confidence intervals of the mean. The dotted red line separates the observation period (i.e., the first 95 matches) from the evaluation period (i.e., the last 5 matches).
(PNG)

**S7 Fig. Trajectories of mean normalised GPM (left panel) and KDA (right panel) for each of six K-means clusters of players with different patterns of match spacing.** Shaded regions indicate 95% confidence intervals. Players in this figure are a subsample who initiate at a similar range of GPM (approximately surrounding the median of the original sample).
(PNG)

**S8 Fig. Trajectories of mean GPM (left panel) and KDA (right panel) for each of eight density-based clusters (HDBSCAN) of players with different patterns of match spacing.** Players in this figure are a subsample who initiate at a similar range of GPM and KDA respectively (approximately surrounding the median of the original sample). Shaded regions indicate 95% confidence intervals.
(PNG)

**S9 Fig. Elbow plot of inertia against number *k* of centroids while running the K-means clustering algorithm.** The dotted vertical red line indicates the point of maximum curvature and thus the selected number of optimal *k* clusters for our K-means clustering of gameplay schedules.
(PNG)

## Acknowledgments

We thank Riot Games for provision of the data set analysed in this work. We would also like to thank Nemanja Vaci, Myat Thura Aung, and Sagarika Patra for valuable discussion.

## Author Contributions

**Conceptualization:** Alex Wade, Tom Stafford.

**Data curation:** Ozan Vardal, Valerio Bonometti, Alex Wade.

**Formal analysis:** Ozan Vardal, Valerio Bonometti.

**Funding acquisition:** Anders Drachen.

**Investigation:** Ozan Vardal.

**Methodology:** Ozan Vardal, Valerio Bonometti, Tom Stafford.

**Resources:** Anders Drachen.

**Supervision:** Anders Drachen, Alex Wade, Tom Stafford.

**Validation:** Valerio Bonometti.

**Visualization:** Ozan Vardal, Valerio Bonometti.

**Writing – original draft:** Ozan Vardal, Valerio Bonometti.

**Writing – review & editing:** Ozan Vardal, Valerio Bonometti, Anders Drachen, Alex Wade, Tom Stafford.

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
