## [Decision Letter · Decision Letter 0]

1 May 2022

PONE-D-22-03288Mind the gap: Distributed practice enhances performance in a MOBA gamePLOS ONE

Dear Dr. Vardal,

Thank you for submitting your manuscript to PLOS ONE. After careful consideration, we feel that it has merit but does not fully meet PLOS ONE’s publication criteria as it currently stands. Therefore, we invite you to submit a revised version of the manuscript that addresses the points raised during the review process.

We look forward to receiving your revised manuscript.

Kind regards,

Rabiu Muazu Musa, PhD

Academic Editor

PLOS ONE

Journal Requirements:

2. In your Methods section, please include additional information about your dataset and ensure that you have included a statement specifying whether the collection and analysis method complied with the terms and conditions for the source of the data.

Reviewers' comments:

Reviewer's Responses to Questions

**Comments to the Author**

1. Is the manuscript technically sound, and do the data support the conclusions?

Reviewer #1: Yes

Reviewer #2: Partly

2. Has the statistical analysis been performed appropriately and rigorously? 

Reviewer #1: Yes

Reviewer #2: Yes

3. Have the authors made all data underlying the findings in their manuscript fully available?

Reviewer #1: No

Reviewer #2: No

4. Is the manuscript presented in an intelligible fashion and written in standard English?

Reviewer #1: Yes

Reviewer #2: Yes

5. Review Comments to the Author

Reviewer #1: ## Summary

The manuscript under review aims to leverage large sample data from the videogame league of legends to assess the influence of distribution of practice through time on performance. They use two performance measures, KDA and GPM that are common metrics across many similar games. They find that distributed practice leads to better performance after 100 games. Further, using various clustering tools to identify similar distributions of games across learning, they find that differences in kinds of game distributions, while overall the same length of time, led to statistically significant but clinically insignificant differences in performance.

## Review

There are many things about the study that are laudable. First, the authors are in the forefront of leveraging large samples of game data to investigate cognition. This work is important because it encourages breadth in cognitive theory by extending relevant data from the simple lab task to complex, real world domains and performance spanning much larger learning trajectories. Second, the work is a model for careful scientific study of complex datasets such as theirs. As task and data complexity goes up, the opportunity for mistakes also increases. Throughout, I found the analysis thoughtful and thorough: it was especially careful about testing possible confounds and artifacts and left the reader with a sense of confidence that findings were robust (if small at times). Third, I found the paper, at all points, admirably well written and clear. Finally, I found the conclusions suitably cautious: despite have a sample that 1,000 time larger than most studies the authors resisted overselling their findings.

I have only one real worry. It is not likely to be the case, but as it impacts the main findings, I think it is worth ruling out. GPM and KDA will vary by position, with support positions getting lower numbers over all. It is also true, anecdotally, that players are encouraged to play support positions when they are new, so that their poor farming speed does not unduly impact their teams chance of winning. What if the extra performance of the distributed group is due to the fact that more of those players are transitioning to core roles than in the groups that mass their play into just a short period? The results would look like what you got, but be caused by differential position change, rather than distributed practice. It should be easy to check the frequency of position in each group. If they similar, then I think you are fine and the work is publishable. (although, now that I think about it, a null effect should also be publishable especially with a large n and a real world complex task.)

I have some notes below on little things that stuck out to me during my reading, but they are minimal, and I think the work is publishable with very minor revisions, assuming my worry above is disproven.

Well done, and thank you for a blessedly easy manuscript review.

## Notes

As League of Legends developer Riot Games 309 keeps the MMR algorithm confidential, we normalised all values of MMR across the 310 data and analyses reported here.

> I’m not sure I understand why normalization is a solution to confidential algorithms.

332 of differences arising prior experience

> typo

Specifically, we expected new players to 346 suffer more losses against the relatively more experienced majority (unobserved in the 347 sample) towards the start of the season, where the matchmaking algorithm has begun 348 to calibrate for fair matches. This intuition is supported by the trajectory of loss 349 percentage, which descends to 50% as the average rating of the sample stabilises 350 (plotted together with MMR).

> this is a nice point, and shows the authors are thinking carefully about the nature of the game, and of the nature of competitive online matches with matchmaking. Displaying some thorough data analysis awareness here.

4.1 Optimising training for MOBAs

> The final advice for players 4.1 (line 616) is not really useful. In paragraph 1 the authors themselves give both practical and reasons of preference to ignore it. The second paragraph has literally nothing to do with the actual findings of the study, and so seems an awkward bit of advice, and a poor fit for the concluding paragraph of the manuscript. While I do really like giving the game community some payback, this might be better cut. If you do cut it, make sure to grab the related sentence in the abstract as well.

Reviewer #2: In the present work, the authors aim at studying the effect of practice distribution on performance in League of Legends, a well-known MOBA game, by extending previous work via exploratory data analysis and statistical analysis. They also claim to investigate through the lens of machine learning whether the timing of breaks influences performance in the game.

Overall, the paper is well-written (with few exceptions) and easy to follow. The authors do a good job in introducing the problem, the relevant literature and discussing some of the gaps in studying how the distribution of practice affects performance. However, there are some major shortcomings that make the present work not ready for being published.

1. In the contributions, the authors state that they aim at “generalising laboratory work to an ecologically valid motor skill environment”. While this point only relates to the fact that League of Legends is used to analyse distribution of practice and its relation to performance, it is not clear how this work generalises previous work done in a laboratory setting in a different way than previous studies on games have done. In particular, the authors highlight how comparisons with laboratory study and new studies are particularly difficult. One specific reason mentioned is the very definition used for practice distribution, which in the first case is the time elapsed between trials, and in the second case is the time gap between the first and last game session. The authors argue that the conflation between practice distribution and frequency is a concern. However, they do use frequency in their very own analysis, thus, limiting their contributions and making their application an extension of previous research on a different type of game. It would have been interesting to actually filling the gap between the two lines of research by comparing different definitions of practice distribution and discussing their generalisability to different contexts.

2. League of Legends is a particular kind of game, which entails team collaboration and role-based playing. Champions are a fundamental part of the game, having their own sets of skills and powers. As the authors also note, champions have different roles, which influence the final performance metrics (used in the paper) that players achieve: a role as a support entails less kills and gold for instance. However, this aspect is not considered in the analysis, where players are evaluated independently of the role and champion they were playing. While looking at each and every champion might have included too much noise in the analysis and led to inconclusive results, the analysis of performance achieved separated by main roles in a team would have made the results stronger in my opinion.

3. In contrast to the data that is available through the API, the authors have access to the MMR which is another interesting metric for performance. However, the authors do not take this aspect into account in the main part of the manuscript. As they notice that the MMR has an opposite trend than the other performance metrics used, they quickly leave this aspect aside. However, this aspect also takes a big part of the discussion at the end of the manuscript, where additional results (in the SI) are introduced. I would suggest at the very least to introduce this discussion beforehand, as having it only at the very end is confusing for the reader.

4. One main concern in the current analysis is related to the use of only ranked games to compute the practice distribution. The authors highlight the fact that this can have an effect on the starting performance that is observed across players, however, it is also not clear how it could affect the results in case players would play other types of matches between ranked sessions.

5. Another weakness in the analysis is related to the samples used and their sizes. Not only the authors end up not considering almost two thirds of the players (as they state In the discussion) but they are also not clear on the samples they use at each step of the analysis and their sizes. In particular, the authors subsample on the basis of the initial GPM and KDA and report a sample size of more than 52K in the first case but none in the second. Moreover, when discussing the results the samples are reduced to a couple of thousands players and in one case to less than 400. However, when reporting values in Table 3, sample sizes seem to be higher. Also, when discussing the time gap (Section 3.1) they use a sample of more than 162K players. The selection process is obscure, and it should be clearly explained.

6. Connected to the previous point, the authors also select three ranges of days for practice, i.e. 1-15, 76-90, 136-150. These ranges are not motivated anywhere. The authors do not provide any background information on the underlying distribution of days, and do not describe how they defined these ranges and why. Moreover, they are not consistent as they always talk about days, but display hours in Figure 3 for instance. I suggest the authors to revise these points to make clear the process behind their data selection and sampling.

7. The discussion around the results in Figure 5 and 6 is not clear. The authors talk about cluster 1 and 4 being the extremes of a spacing spectrum. However, it seems that cluster 1 and 3 should be mentioned instead. Moreover, they describe Figure 6 as the temporal distribution of matches, and how clusters are characterised by different timing and intensity. However, this discussion seems related to the second subplot of Figure 5 as Figure 6 displays the performance metrics in time spliced across clusters.

8. Finally, the figures could be improved by using different line styles. At the moment colours are the only element used to distinguish among the lines which are close and overlapping. Moreover, even if Figure 3 has the distributions along the two axes displayed, it is still hard to understand the density of the points in the scatter plot. A better visualisation could make the use of a heat-map, for instance, to clearly show the two-dimensional distribution of the data.

Minor comment: I recommend language editing, as there are a few typos.

---

## [Author Response · Author response to Decision Letter 0]

23 Aug 2022

PONE-D-22-03288 Revision

Ozan Vardal

Department of Computer Science, University of York

17th May 2022

Dr Musa

 Attached is our revision of “Mind the gap: Distributed practice enhances performance in a MOBA game” [PONE-D-22-03288]. We thank the reviewers for their substantive feedback and helpful comments. We have made a number of changes to the submission including further analysis and amendments to figures and text, which we believe reflect the concerns of you and your reviewers. We describe these changes below, commencing with a joint area of concern raised by both reviewers, followed by a point by point discussion of issues raised by individual reviewers.

 As before, all analysis code, including the present changes, can be viewed at the following repository: https://github.com/ozvar/lol_practice_distribution. We look forward to further discussion of our paper.

Kind regards,

Ozan Vardal, Valerio Bonometti, Anders Drachen, Alex Wade, Tom Stafford

RE: GPM/KDA will vary by position. An analysis of performance should therefore account for position played.

 We thank both reviewers for noting that our analysis of performance could benefit from accounting for the position of the champion that was played. Notably, R1 raises an interesting point that the trajectory of performance exhibited by our spacing group (i.e., depressed initial gains followed by superior final performance) could possibly be explained by players in this group starting their games in the support role, before transitioning to core roles later on.

 In order to test this hypothesis we visualised the trajectory of the proportion of matches played in the support role at each nth match (see figure below in attached document), for each of our spacing groups and for each of our subsamples matched on initial performance (left panel GPM-matched, right panel KDA-matched). Contrary to this hypothesis, we note that players in the group that spaced their first and 95th match the most have the lowest proportion of support games played, but this proportion is consistently low. Thus, our result does not appear to be explained by players transitioning from low earning support roles to high earning core roles. Nevertheless we agree that our analysis is strengthened by devoting additional attention to this aspect of the dataset. 

RE: Language edits

 Following suggestions by both reviewers and a general proofread, we have corrected several typos including the one suggested by R1 (line 332 in the original manuscript).

Reviewer 1

RE: The final advice for players 4.1 (line 616) is not really useful. In paragraph 1 the authors themselves give both practical and reasons of preference to ignore it.

 We have addressed this concern by cutting the mentioned paragraph from the discussion and also the related clause from the abstract. Instead of concluding with policy advice for players, we have ended with a more traditional conclusion section summarising our key findings as well as our recommendations for follow-up research:

 Research on motor learning has demonstrated that taking breaks between practice sessions, as opposed to massing them in relatively short time windows, benefits ultimate performance [24,25]. By analysing an observational, longitudinal data set describing player performance in a massive, commercially successful video game, we showed that the distributed practice effect is relevant in an ecologically valid context comprising stakeholders with a vested interested in improving their skills. Although data sets such as ours afford strong statistical power and the ability to filter through observations that meet desired experimental conditions, they are also complicated by noise and potential confounds. As a solution, we propose that researchers seeking to use telemetry data adopt a hybrid approach, collecting demographic information on players before tracking their play records through game APIs. In doing so, interested researchers may control for variables related to initial performance, such as age or cognitive ability [70], and account for sources of data pollution such as players generating data on multiple accounts.

Reviewer 2

RE: Our aim of “generalising laboratory work to an ecologically valid motor skill environment” was not satisfied due to the conflation of practice distribution with frequency

 We thank R2 for raising an interesting discussion point regarding the generalisability of our findings. We concede that our work is not a direct translation of past laboratory experiments due to the way we operationalised practice distribution. We add that there is some generalisation, as time between individual trials and time between first and last trial is typically correlated. 

 To clarify this discrepancy, we have amended the initial declaration of aims in our introduction section to the following:

"In the current study we extended this line of enquiry to a popular commercial action

game, with the aim of generalising work on distributed practice that has been

conducted using artificial tasks created by researchers, to a non-artificial, ecologically

valid environment with which researchers have not interfered."

We have additionally added the following paragraph in our discussion section to make this limitation clear:

"In our case, the absence of experimental control restricted our ability to compare groups of players with homogenous time gaps between each of their play sessions, as has been done in laboratory studies of distributed practice [23]. Our solution, similar to other studies that have used game telemetry [31, 32, 36] was to use a proxy for intersession time interval, namely the time gap between the first and last match. Although time between first and last match is likely related to time between individual trials, we acknowledge that use of this alternative operationalisation limits our ability to generalise from laboratory work to a non-artificial environment."

RE: Introducing additional (SI) results on MMR in the discussion is confusing

We note R2’s concern regarding our handling of MMR-related results in the discussion section. To reiterate, our issue with this metric is that the way it is calculated is unknown to anyone besides Riot Games. Although we are aware it is somewhat dependent on match outcome, match outcome itself is dependent on many factors including the behaviour of teammates and opponents. We believe this makes the metric less fit for our purposes than GPM and KDA, which relate more clearly to the performance of the individual at each match. 

In light of these reflections we have decided to abridge the paragraph on MMR in the discussion section. To eliminate any related confusion, we have prefaced both the plot (S3 Fig) introduced in the discussion section as well as our rationale behind discontinuing the analysis of MMR earlier on in the results section as suggested.

RE: Our sample exclusively contains ranked matches. How might players playing unranked matches between ranked ones affect the results?

 This is an important point which we acknowledge has not been adequately addressed in our manuscript. We were indeed agnostic to any experience that players may have acquired in unranked game modes that were engaged with between matched that were recorded in our data set. Relatedly, it might be possible that players who spaced their games the most may have been playing unranked matches in between the ranked matches that we observed. 

 We believe that our results do not support this hypothesis, as we would then have expected the players that spaced their matches the most to have a more accelerated learning curve than what we observed (Figure 2), reflecting the additional practice hours that they accumulated. Instead we observed that players who spaced their matches the most started with a comparatively depressed learning curve. We have included several sentences in the discussion section to make note of this possibility.

RE: Missing information on sample sizes in text

 We thank R2 for bringing the absence of vital analysis information to our attention. Firstly, we have added a paragraph at the end of 2.3 clearly explaining the data preprocessing steps that led to us focusing our analysis on a third of the original player sample. Secondly, we have amended the text describing our statistical analysis of KDA so that it mentions the size of that particular subsample. Finally, while checking that each portion of the analysis contains clear descriptions of sample sizes, we noticed that our regression analysis does, and in fact was erroneously conducted on the entire sample as opposed to our subsamples matched on initial performance. We have corrected this portion of the analysis together with all in-text descriptions of results, ensuring that sample sizes are mentioned in all cases.

RE: Variations in sample sizes across analyses are not clear

 We would like to clarify the reason for our varying sample sizes: Our analysis is split into two sections. In the first section we grouped players based on their practice schedules using a rule-based approach inspired by previous research. These groups, which we view as the observational analogue to experimental conditions, were extracted from our pre-processed sample of 162k players. Recognising a limitation in our rule-based subsampling, we conducted a novel exploratory analysis in the second section, this time grouping different practice schedules using a data-driven, machine learning approach. As we wished to compare this second, alternative method with our first approach, we performed our clustering analysis on the pre-processed sample of 162k. We have added some text to the first paragraph of 3.1 to make this clearer.

 In both sections we are selecting groups of players that meet our “experimental criteria” (e.g.., massing versus spacing practice), which relates to play behaviour that is atypical of the larger population. Subsampling players engaging in atypical behaviour naturally reduces our group sizes. Further, having recognised that the correlation between initial ability and play intensity confounds our analysis, in both sections of our analysis engaged in additional filtering, subsampling players on initial ability (based on a range surrounding the median performance) prior to statistical analysis. This resulted in further reduction to our group sizes. 

In sum, as our research question relates to particular groups of players who behave in specific ways, we argue that it is not unusual for the sizes of groups in our analysis to be a small fraction of the entire sample. Instead, they are the result of a series of theoretically motivated filtering steps. We have taken steps to ensure that these filtering steps are clearly described in paragraphs 2 and 3 of the Results section (see also rebuttal point below on subsampling players by day range).

 The example R2 provides of a group numbering less than 400 is a result of these same filtering steps: 1) following preprocessing we retained ~162k players, 2) matching on initial KDA we retained ~17k players, 3) of these players, ~400 played took between 136-150 days to play their first 100 ranked matches. Other groups in our analysis contained more players than this particular group. This relates to the highly skewed distribution of time gap between first and 95th match (see S2 Fig, also below in attached document). 

 It is worth noting that this group of ~400 players and the other groups in our statistical analyses are not related to the groups displayed in Table 3. This table serves only to describe the number of players that were shared between each pair of groups arising from the first (rule-based clustering) and second (ML clustering) section of our analysis. We did not conduct any statistical analyses using groups of sizes displayed in Table 3, it is a descriptive table of joint frequencies only. To make this clearer, we have extended the caption for Table 3. We trust that these steps coupled with the explanations provided above clarify the discrepancies relating to our sample sizes.

RE: The three ranges of days used to subsample players should be clearly motivated in text

 The day ranges used to subsample players were motivated by visualising the frequency distribution of the time elapsed in days between the first and last match for each player (S2 Fig, below), and choosing three groups that were sufficiently discrete from one another and that contained adequate sample sizes for analysis. We have addressed R2’s concern by including this previously absent description in the Results section and referencing the figure.

RE: Figure 3 displays time gap in hours, but text refers to time gap in days

 The x axis in Figure 3 should indeed be labelled as time gap in days. We have corrected the x axis label and made sure the x axis metric is clearly described in the figure caption. 

 Plots depicting our machine learning clusters refer to time gap in hours, not days, as players were clustered based on their time series of inter-match time gap We have checked to ensure that this is clearly described in the section describe our clustering, and included a description of the time gap metric in every relevant figure caption for our clustering analysis (Figure 4, Figure 5, S5 Fig, S6 Fig).

RE: Discussion around figures relating to clustering analyses could be improved

 We thank R2 for raising important points surrounding the discussion of our clustering analysis in section 3.1. Our discussion of how matches were temporally distributed does indeed center on figure 5, not 6. A further mistake involves our describing cluster 4 as the group of players with the earliest and shortest rest period, when the text should actually refer to cluster 3 as suggested. We have corrected these two mistakes, and also restructured the remainder of the paragraphs describing our clustering analysis to make the section clearer overall.

RE: Suggestions regarding improvements to figures

 We agree with R2 that density is difficult to discern the density of points in Figure 3. We suggest that a scatter plot may be better at displaying the relationship between our two variables than the proposed heat map, but have elected as a middle-ground option to reduce the size and opacity of our points in our scatterplot to make density more apparent.

---

## [Decision Letter · Decision Letter 1]

13 Sep 2022

PONE-D-22-03288R1Mind the gap: Distributed practice enhances performance in a MOBA gamePLOS ONE

Dear Dr. Vardal,

Thank you for submitting your manuscript to PLOS ONE. After careful consideration, we feel that it has merit but does not fully meet PLOS ONE’s publication criteria as it currently stands. Therefore, we invite you to submit a revised version of the manuscript that addresses the points raised during the review process.

We look forward to receiving your revised manuscript.

Kind regards,

Rabiu Muazu Musa, PhD

Academic Editor

PLOS ONE

Journal Requirements:

Reviewers' comments:

Reviewer's Responses to Questions

**Comments to the Author**

1. If the authors have adequately addressed your comments raised in a previous round of review and you feel that this manuscript is now acceptable for publication, you may indicate that here to bypass the “Comments to the Author” section, enter your conflict of interest statement in the “Confidential to Editor” section, and submit your "Accept" recommendation.

Reviewer #1: All comments have been addressed

Reviewer #2: (No Response)

2. Is the manuscript technically sound, and do the data support the conclusions?

Reviewer #1: Yes

Reviewer #2: Yes

3. Has the statistical analysis been performed appropriately and rigorously? 

Reviewer #1: Yes

Reviewer #2: Yes

4. Have the authors made all data underlying the findings in their manuscript fully available?

Reviewer #1: Yes

Reviewer #2: No

5. Is the manuscript presented in an intelligible fashion and written in standard English?

Reviewer #1: Yes

Reviewer #2: Yes

6. Review Comments to the Author

Reviewer #1: I have no further concerns, and believe the research is now suitable for publication. I thank the authors once again for an interesting read.

Reviewer #2: I thank the authors for addressing my concerns and providing, in particular, additional evidence of the effects of role playing on the group performance results. Overall, I believe that their revisions make the present work almost ready to be published.

I only have two final (and minor) edits:

- When replying to my comment about the MMR, the authors also notice that "Although [MMR] is somewhat dependent on match outcome, match outcome itself is dependent on many factors including the behaviour of teammates and opponents." I think the authors should actually add this observation to the discussion about MMR in the paper.

- In the edited text about ranked/unranked matches the authors write "our results 2". Should this be Figure 2?

7. PLOS authors have the option to publish the peer review history of their article (what does this mean?). If published, this will include your full peer review and any attached files.

Reviewer #1: No

Reviewer #2: No

---

## [Author Response · Author response to Decision Letter 1]

20 Sep 2022

PONE-D-22-03288 Second Revision

Ozan Vardal

Department of Computer Science, University of York

20th September 2022

Dr Musa

Attached is our second revision of “Mind the gap: Distributed practice enhances performance in a MOBA game” [PONE-D-22-03288]. We thank the editor and reviewers for their careful inspection of our previous rebuttal and for leaving their final comments. We have made several changes in light of your latest responses, which we outline below.

We look forward to further updates on our paper.

Kind regards,

Ozan Vardal, Valerio Bonometti, Anders Drachen, Alex Wade, Tom Stafford

RE: Please review your reference list [...]

In line with the specified journal requirements for mentioning any changes to our reference list, we would like to clarify the following changes to our references:

References to MacQueen et al. (1967), Tavenard et al. (2020), Campello et al. (2013), and McInnes et al. (2017) were remnants of a previous iteration where appendix methods existed in the main body text. These citations no longer exist in the main body manuscript and have therefore been removed from the reference list.

In response to initial reviews, we also made significant changes to our concluding paragraphs. One paragraph including references to Baddeley and Longman (1978) and Schmidt and Lee (2019) was removed, resulting in removal of these papers from our reference list. In reframing our final paragraph, we included a supporting reference to Kokkinakis et al. (2017), which introduces readers to relevant work relating to potentially confounding variables that we mention in our discussion, and that are in our domain of interest.

RE: When replying to my comment about the MMR, the authors also notice that "Although [MMR] is somewhat dependent on match outcome, match outcome itself is dependent on many factors including the behaviour of teammates and opponents." I think the authors should actually add this observation to the discussion about MMR in the paper.

We agree with Reviewer 2 that this is helpful clarification for the reader. We have expanded our paragraph in the Discussion section on MMR to state: 

Moreover, although MMR is partly dependent on match outcome, the probability of winning a match is dependent on many factors (including the behaviour of teammates and opponents), and is itself the subject of many efforts in prediction. 

RE: In the edited text about ranked/unranked matches the authors write "our results 2". Should this be Figure 2?

We thank Reviewer 2 for pointing out this confusing sentence, which is indeed intended to reference Figure 2. We have amended it to “[...] than what is observed in Figure 2 [...]”.

---

## [Editor Report · Decision Letter 2]

26 Sep 2022

Mind the gap: Distributed practice enhances performance in a MOBA game

PONE-D-22-03288R2

Dear Dr. Vardal,

We’re pleased to inform you that your manuscript has been judged scientifically suitable for publication and will be formally accepted for publication once it meets all outstanding technical requirements.

Kind regards,

Rabiu Muazu Musa, PhD

Academic Editor

PLOS ONE
---

## [Editor Report · Acceptance letter]

6 Oct 2022

PONE-D-22-03288R2 

Mind the gap: Distributed practice enhances performance in a MOBA game 

Dear Dr. Vardal:

I'm pleased to inform you that your manuscript has been deemed suitable for publication in PLOS ONE. Congratulations! Your manuscript is now with our production department. 

Kind regards, 

on behalf of

Dr. Rabiu Muazu Musa 

Academic Editor

PLOS ONE